# ED-NeRF: Efficient Text-Guided Editing of 3D Scene With Latent Space NeRF

**Jangho Park**[2][*] **Gihyun Kwon**[3][*], **Jong Chul Ye**[1,2,3]
Kim Jaechul Graduate School of AI[1], Robotics Program[2],
Department of Bio and Brain Engineering[3], KAIST
{jhq1234,cyclomon,jong.ye}@kaist.ac.kr

## Abstract

Recently, there has been a significant advancement in text-to-image diffusion models, leading to groundbreaking performance in 2D image generation. These advancements have been extended to 3D models, enabling the generation of novel 3D objects from textual descriptions. This has evolved into NeRF editing methods, which allow the manipulation of existing 3D objects through textual conditioning. However, existing NeRF editing techniques have faced limitations in their performance due to slow training speeds and the use of loss functions that do not adequately consider editing. To address this, here we present a novel 3D NeRF editing approach dubbed ED-NeRF by successfully embedding real-world scenes into the latent space of the latent diffusion model (LDM) through a unique refinement layer. This approach enables us to obtain a NeRF backbone that is not only faster but also more amenable to editing compared to traditional image space NeRF editing. Furthermore, we propose an improved loss function tailored for editing by migrating the delta denoising score (DDS) distillation loss, originally used in 2D image editing to the three-dimensional domain. This novel loss function surpasses the well-known score distillation sampling (SDS) loss in terms of suitability for editing purposes. Our experimental results demonstrate that ED-NeRF achieves faster editing speed while producing improved output quality compared to state-of-the-art 3D editing models. Code and rendering results are available at our project page[1].

## 1 Introduction

In recent years, the development of neural implicit representation for embedding three-dimensional images in neural networks has seen remarkable progress. This advancement has made it possible to render images from all angles using only a limited set of training viewpoints. Starting with the seminar work known as the Neural Radiance Field (NeRF) (Mildenhall et al., 2021), which trained radiance fields using a simple MLP network, various improved techniques (Barron et al., 2021; Reiser et al., 2021; Müller et al., 2022) based on advanced network architectures or modified encoding have been proposed. Alternatively, several methods (Sun et al., 2022; Fridovich-Keil et al., 2022; Karnewar et al., 2022; Chen et al., 2022) proposed to directly optimize voxel points serving as sources for rendering, bypassing the traditional approach of encapsulating all information within implicit networks. These methods have gained prominence for their ability to train radiance fields in a remarkably short time. In addition to representing existing 2D image data in the 3D space, recent research has explored expanded approaches for generating entirely novel 3D objects. With the emergence of text-to-image embedding models like CLIP (Radford et al., 2021), various methods have been proposed to train implicit networks that can generate new objects solely from text prompts (Jain et al., 2022). This trend has been accelerated with the advent of text-to-image diffusion generation models such as Stable Diffusion (Rombach et al., 2022), particularly through the score distillation sampling (SDS) (Poole et al., 2022) which conveys the representation of the text-to-image model to NeRF model.

---

[*]equally contributed
[1]https://jhq1234.github.io/ed-nerf.github.io/

NeRF Scene               Edited novel scene

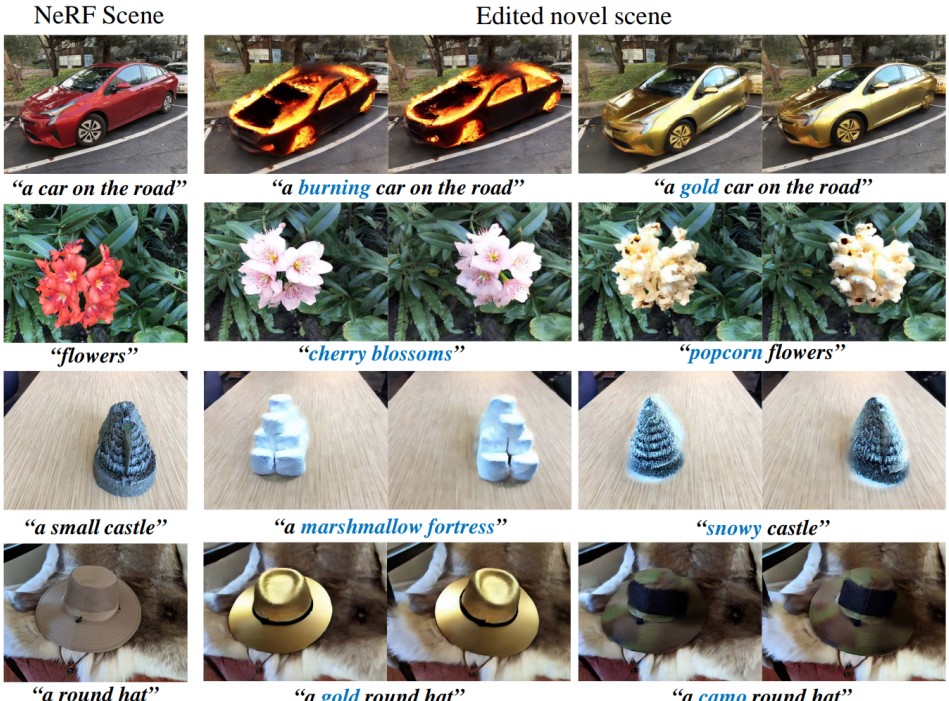

Figure 1: **Qualitative results of our method.** ED-NeRF successfully edited 3D scenes with given target text prompts while preserving the original object structure and background regions.

However, the challenge of editing pre-trained 3D implicit networks according to specific conditions still remains as an open problem due to the constraints of tasks: maintaining the integrity of the original 3D images while making desired modifications. As an initial work, several approaches (Wang et al., 2022; 2023a) tried to edit the pre-trained NeRF models based on text conditions, utilizing the pre-trained CLIP model to fine-tune the parameters of NeRF models. Nevertheless, these methods exhibit notable weaknesses, including the performance limitations of the CLIP model itself and the need for rendering high-resolution images during training, which results in significant time consumption.

Recently, several editing methods proposed to leverage the enhanced expressiveness of text-to-image diffusion models such as Stable Diffusion. Some methods (Sella et al., 2023) proposed to directly employ the score distillation sampling method, with additional regularizations. However, these methods suffer from significant time consumption and instability in generation performance due to the requirement of full-resolution rendering in the training stage and limitations of the score distillation loss itself. Other alternative approaches (Haque et al., 2023) proposed to directly manipulate the training images of NeRF using text-guided image translation models. This method aims to enable the generation of 3D images corresponding to text conditions. However, it suffers from a significant drawback in terms of training time, as it requires periodic translation of training images during the training process.

To address these challenges, we are interested in developing a novel NeRF editing method to efficiently and effectively edit 3D scenes using only text prompts. To achieve this, we enable NeRF to operate directly in the NeRF latent space, similar to Latent-NeRF (Metzer et al., 2023), which helps reduce time and computational costs. However, naively rendering the latent feature of real-world scenes directly with NeRF may lead to a significant drop in view synthesis performance due to the lack of geometric consistency in the latent space. To tackle this issue, we conduct an analysis of the latent generation process and propose a novel refinement layer to enhance performance based on the analysis. Furthermore, to solve the drawback of the existing SDS-based method in editing, we propose a new sampling strategy by extending Delta Denoising Score (DDS) (Hertz et al., 2023), a 2D image editing technique based on score distillation sampling, into the 3D domain. This extension allows us to achieve high-performance editing capabilities while keeping computational costs affordable, even with large Diffusion Models such as Stable Diffusion. Given the superior editing proficiency of our approach, we've named it ED-NeRF (EDiting NeRF).

## 2 RELATED WORK

Starting from the Neural Radiance Field (NeRF) (Mildenhall et al., 2021), there have been approaches to represent three-dimensional scenes in neural fields. However, due to the slow training speed, several approaches tried to improve the performance by modifying the network architecture or training strategy (Barron et al., 2021; Müller et al., 2022; Reiser et al., 2021). Several methods without relying on neural networks showed great performance in accelerating. These include a method for optimizing the voxel fields (Sun et al., 2022; Fridovich-Keil et al., 2022; Chen et al., 2022; Karnewar et al., 2022), or decomposing the components of field representation. Based on the success of these techniques, methods for generating 'novel' 3D scenes have been proposed. Especially with the emergence of the text-to-image embedding model of CLIP (Radford et al., 2021), DreamField (Jain et al., 2022) leveraged CLIP to train the NeRF model for novel 3D object synthesis. Recently, the performance of the text-to-image diffusion model enabled remarkable improvement in 3D generation. Starting from DreamFusion (Poole et al., 2022), several methods (Metzer et al., 2023; Liu et al., 2023b; Xu et al., 2023) showed impactful results using the diffusion-based prior. However, these methods are limited to generating 'novel' 3D objects and, therefore cannot be applied to our case of NeRF-editing which tries to modify the existing 3D scenes according to the given conditions.

Compared to the novel object generation, NeRF editing is still not an explored field, due to the complexity of the task. As a basic work, several methods focused on color or geometric editing (Yuan et al., 2022; Liu et al., 2021; Kuang et al., 2023). Other works tried style transfer or appearance transfer on 3D neural fields (Zhang et al., 2022; Liu et al., 2023a; Bao et al., 2023) and showed promising results. With incorporating the CLIP model, several approaches (Wang et al., 2022; 2023a; Song et al., 2023) tried to modify the pre-trained NeRF towards the given text conditions. Although the results show pleasing results, the method still has limitations in detailed expression due to the limitation of CLIP model itself.

Similar to the novel scene generation case, the development of text-to-image diffusion models brought significant improvement in the editing field. Starting from Score Distillation Sampling method proposed in DreamFusion, Vox-e tried to edit the pre-trained voxel fields with regularization (Sella et al., 2023). As an alternative method, InstructNerf2Nerf (Haque et al., 2023) proposed to directly leverage 2D image translation models for changing the attribute of 2D images for NeRF training. However, these methods still have limitations due to excessive training time or unstable editing from loss functions. To address the above problems, we propose an efficient method of editing with novel latent space NeRF training and improved edit-friendly loss functions.

## 3 METHODS

Figure 2 provides an overview of training ED-NeRF. First, we optimize NeRF in the latent space of Stable Diffusion. To do this, we encode all images using a pre-trained Variational Autoencoder (VAE) to obtain the feature vectors and guide NeRF to predict these feature vectors directly. Also, we introduce an additional refinement layer, which enhances the novel view synthesis performance of NeRF (Fig. 2(a)). At the inference stage, we can render a natural image by latent NeRF via decoding rendered latent map (Fig. 2(b)). At the editing phase, by utilizing DDS, we adjust the parameters of both NeRF and the refinement process to align the 3D scene with the provided target text (Figure 3). The detailed pipeline for this approach is outlined in the following sections.

### 3.1 ED-NeRF FOR 3D SCENE EDITING

NeRF (Mildenhall et al., 2021) uses MLPs to predict density $\sigma$ and color $\mathbf{c}$ for a given 3D point coordinate $\mathbf{x} = (x, y, z)$ and view direction $\mathbf{d}$. Through positional encoding $\gamma(\cdot)$, $\mathbf{x}$ and $\mathbf{d}$ are mapped into high-frequency vectors, and then fed into the neural network of NeRF, resulting in two outputs: density $\sigma \in \mathbb{R}$ and color $\mathbf{c} \in \mathbb{R}^3$.

$$(\mathbf{c}, \sigma) = F_\theta^c(\gamma(\mathbf{x}), \gamma(\mathbf{d})) \tag{1}$$

Through volume rendering Eq. (2), NeRF predicts the pixel color along the camera ray $\mathbf{r}(t) = \mathbf{o} + t\mathbf{d}$, with $t$ representing the depth within the range $[t_{near}, t_{far}]$, $\mathbf{o}$ stands for the camera position,

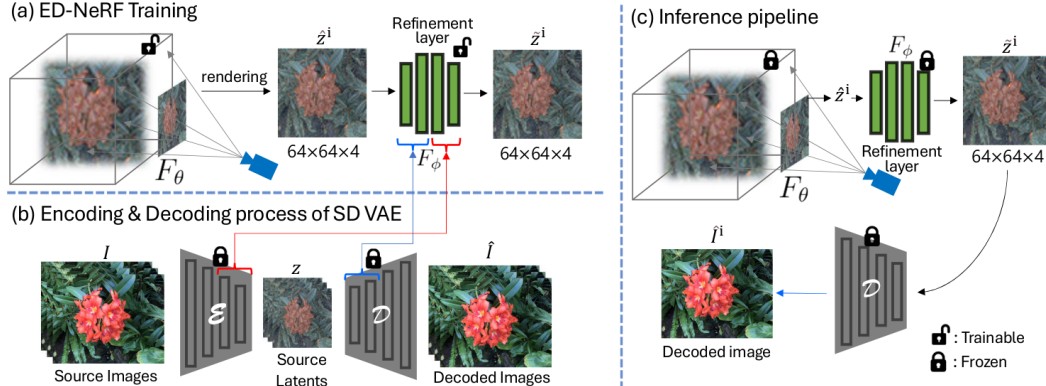

Figure 2: **Overall pipeline of training and inference stage.** (a) We optimize ED-NeRF in the latent space, supervised by source latent. Naively matching NeRF to a latent feature map during optimization can degrade view synthesis quality. (b) Inspired by the embedding process of Stable Diffusion, we integrated additional ResNet blocks and self-attention layers as a refinement layer. (c) All 3D scenes are decoded from the Decoder when ED-NeRF renders a novel view feature map.

and $\mathbf{d}$ represents the view direction:

$$\hat{C}(r) = \int_{t_n}^{t_f} T(t)\sigma(\mathbf{r}(t))\mathbf{c}(\mathbf{r}(t), d)dt, \text{ where } T(t) = \exp\left(-\int_{t_n}^{t} \sigma(\mathbf{r}(s))ds\right). \qquad (2)$$

Optimizing NeRF to render the latent feature value of the latent diffusion model offers several advantages in text-guided 3D generation. These advantages include a reduced training burden due to the decreased dimensionality of the space, and enhanced editability for the NeRF model, as the rendered outputs can be directly employed as input for the latent diffusion models. The concept of migrating NeRF to the latent space is first proposed by Latent-NeRF (Metzer et al., 2023), in which the NeRF is directly trained with the latent feature rather than RGB color. Therefore it can render a 3D scene without the encoding process during optimization when using the latent diffusion model as semantic knowledge prior. However, this work exclusively focuses on generating 'virtual' 3D assets without supervision, making it unsuitable for real-world scenes.

Thus, ED-NeRF is realized based on a novel latent NeRF training pipeline for synthesizing real-world scenes in the latent space. As depicted in Figure 2, when a real-world image dataset $I$ contains multi-view images $I = \{I^i\}_{i=1}^N$, we can encode all images to the latent space of Stable Diffusion via encoder to obtain the feature: $z^i = \mathcal{E}(I^i) \in \mathbb{R}^{64 \times 64 \times 4}$. After embedding all images, we can use the latent feature maps $z := \{z^i\}_{i=1}^N$ as label data set for ED-NeRF training using the loss function:

$$\mathcal{L}_{rec} = \sum_{\mathbf{r} \in \mathcal{R}} \left\| Z^i(\mathbf{r}) - \hat{Z}^i(\mathbf{r}) \right\|^2 \qquad (3)$$

where $Z^i$ denotes the pixel latent value of the latent $z^i$ and $\hat{Z}^i(\mathbf{r})$ is rendered by the volume rendering equation:

$$\hat{Z}^i(\mathbf{r}) = \int_{t_n}^{t_f} T(t)\sigma(\gamma(t))\mathbf{f}_z(\mathbf{r}(t), d)dt, \text{ where } T(t) = \exp\left(-\int_{t_n}^{t} \sigma(\mathbf{r}(s))ds\right). \qquad (4)$$

where $\mathbf{f}_z \in \mathbb{R}^4$ denotes the predicted feature value by the neural network, taking $\gamma(\mathbf{x})$ and $\gamma(\mathbf{d})$ as input:

$$(\mathbf{f}_z, \sigma) = F_\theta(\gamma(\mathbf{x}), \gamma(\mathbf{d})) \qquad (5)$$

By minimizing the loss Eq. (3) to update the parameters of the neural network $F_\theta$, we obtain a novel ED-NeRF model optimized in the latent space of the Stable Diffusion.

### 3.2 REFINEMENT LAYER BASED ON LATENT FEATURE ANALYSIS

When naively matching the latent generated by Eq. (3), we observed that the reconstruction performance significantly deteriorated. In addressing this issue, we analyzed the Encoder $\mathcal{E}$ and Decoder $\mathcal{D}$ of Stable Diffusion and discovered the following insight in the process:

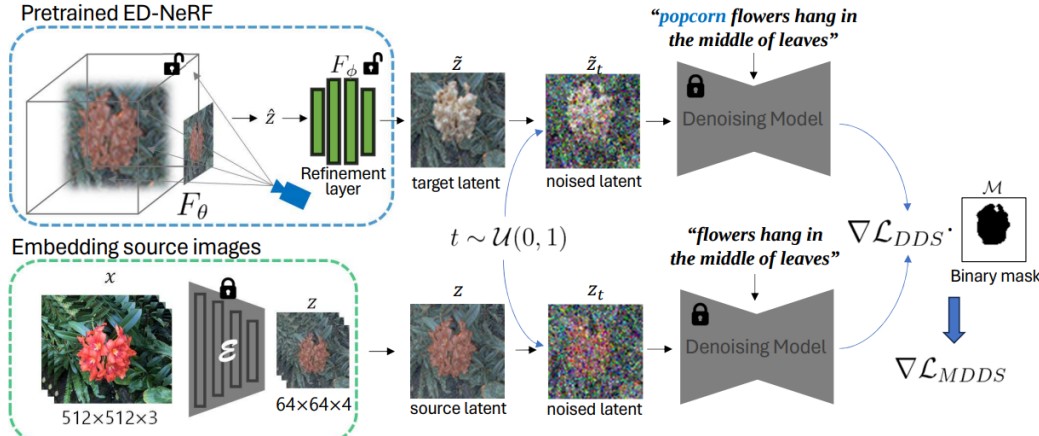

Figure 3: **Expanding DDS into 3D for ED-NeRF editing.** Pretrained ED-NeRF renders the target latent feature map, and a scheduler of the denoising model perturbs it to the sampled time step. Concurrently, the scheduler adds noise to the source latent using the same time step. Each of them is fed into the denoising model, and the DDS is determined by subtracting two different SDS scores. In combination with a binary mask, masked DDS guides NeRF in the intended direction of the target prompt without causing unintended deformations.

1) The encoder and decoder consist of ResNet blocks and self-attention layers. Therefore during the process of mapping the image to the latent space and forming a feature map, pixel values exhibit interference between each other, primarily due to ResNet and self-attention layers. Thus the latent and image pixels are not directly aligned.

2) When NeRF renders a single pixel value from the latent feature map, each ray independently passes through an MLP to determine the pixel value of the feature map. Therefore, the feature value rendered by NeRF for a single pixel is determined without interactions with other pixels.

Based on this analysis, we find that the reason for the deformed reconstruction performance of the latent NeRF lies in the inconsideration of the interactions mentioned above. Therefore, we aim to incorporate the interactions among pixels introduced by the ResNet and self-attention layers into the ED-NeRF rendering stage. Fortunately, in the Encoder and Decoder of Stable Diffusion, the embedded feature maps pass through self-attention layers at the same dimension, allowing us to concatenate two attention layers straightly. Taking advantage of this, we can design a refinement layer $F_\phi(\cdot)$ as shown in Figure 2, without dimension change of input and output vector. Let $\tilde{Z}^i$ as the pixel latent vector of the refined feature map $\tilde{z}^i$, where formed from $\tilde{z}^i = F_\phi(\hat{z}^i)$. Therefore, we can design a refined reconstruction loss function as follows :

$$\mathcal{L}_{ref} = \sum_{\mathbf{r} \in \mathcal{R}} \left\| Z^i(\mathbf{r}) - \tilde{Z}^i(\mathbf{r}) \right\|^2 \text{ , where } \tilde{z}^i = F_\phi(\hat{z}^i) \tag{6}$$

Ultimately, we can formulate total training loss as the sum of the refinement loss $\mathcal{L}_{ref}$ and reconstruction loss $\mathcal{L}_{rec}$, as follows.

$$\mathcal{L}_{rtot} = \lambda_{rec}\mathcal{L}_{rec} + \lambda_{ref}\mathcal{L}_{ref} \tag{7}$$

We update NeRF and refinement layer concurrently denoted as $F_\theta$ and $F_\phi$ by minimizing total loss $\mathcal{L}_{rtot}$ to reconstruct latent vectors in various views. To ensure stable learning, training with $\lambda_{rec}$ set to 1.0 and $\lambda_{ref}$ set to 0.1 during the initial stages of training. Beyond a specific iteration threshold, we set it to 0 to encourage the refinement layer to focus more on matching the latent representations.

### 3.3 EDITING ED-NeRF VIA DELTA DENOISING SCORE

After optimizing ED-NeRF in the latent space, it is possible to directly employ the latent diffusion model to update ED-NeRF parameter via rendered latent map $z$ in the direction of the target text prompt $y_{trg}$. The most well-known method for text-guided NeRF update is Score Distillation Sampling (SDS), which directly transfers the score estimation output as a gradient of NeRF training:

$$\nabla_\theta \mathcal{L}_{\text{SDS}}(\mathbf{z}, y_{trg}, \epsilon, t) = \omega(t)(\epsilon_\psi(\mathbf{z_t}, y_{trg}, t) - \epsilon)\frac{\partial \mathbf{z_t}}{\partial \theta} \tag{8}$$

However, in our NeRF editing case, the updating rule for SDS often shows several problems including color saturation and mode-seeking (Wang et al., 2023b). We conjecture that the problem originated from the properties of score estimation itself. Since the target noise $\epsilon$ is pure Gaussian, the score difference is not aware of any prior knowledge of source images. Therefore the generated outputs are just the replacement of hallucinated objects without consideration of source NeRF.

To solve the problem of SDS, we focus on the recently proposed 2D editing method of Delta Denoising Score (DDS) (Hertz et al., 2023). The major difference between SDS and DDS is that the distilled score is the difference between the denoising scores from target and source. As shown in Eq. (9), DDS can be formed as a difference between two SDS scores conditioned on two different text prompts:

$$\nabla_\theta \mathcal{L}_{\text{DDS}} = \nabla_\theta \mathcal{L}_{\text{SDS}}(\hat{\mathbf{z}}, y_{trg}) - \nabla_\theta \mathcal{L}_{\text{SDS}}(\mathbf{z}, y_{src}), \tag{9}$$

where $\mathbf{z}$ is source latent, $\hat{\mathbf{z}}$ is rendered target latent, $y_{trg}$ represents the target text embedding, $y_{src}$ represents the reference text embedding. DDS guides the optimized latent towards the target prompt from the source prompt without the influence of the pure noise component, therefore it can easily edit 2D images.

We aim to extend this manipulation capability of DDS to 3D space as shown in Fig. 3. As we already have embedded source latent $z^i$ for the $i$-th camera pose, we can directly use them as source components of DDS. To fine-tune the model, we render the edited output $\tilde{z}^i$ which is also rendered from the $i$-th camera pose. With the paired latents, we add the same sampled noise $\epsilon_t$ with the noise scale of timestep $t$ to both source and edited latents so that we obtain noisy latent $\tilde{z}_t^i, z_t^i$. Then we apply the diffusion model to obtain estimated score outputs from noisy latents using different text conditions for source and edited images. As in Eq. (9), we can use the difference between the two outputs as a gradient for updating the NeRF parameters. In this step, we simultaneously train the NeRF parameters $\theta$ with refinement parameters $\phi$ as it showed better editing quality. Therefore with the random $i$-th camera pose, our 3D DDS is formulated as:

$$\nabla_{\theta,\phi} \mathcal{L}_{\text{DDS}} = \nabla_{\theta,\phi} \mathcal{L}_{\text{SDS}}(\tilde{\mathbf{z}}^i, y_{trg}) - \nabla_{\theta,\phi} \mathcal{L}_{\text{SDS}}(\mathbf{z}^i, y_{src}). \tag{10}$$

Although the DDS formulation improves the performance, using vanilla DDS leads to excessive changes in unwanted areas and inconsistency between two different scenes. Therefore, we propose an additional binary mask for utilizing DDS in 3D images. The objective function that combines the binary mask $\mathcal{M}$ and DDS is as follows:

$$\nabla_{\theta,\phi} \mathcal{L}_{\text{MDDS}} = \mathcal{M} \cdot (\nabla_{\theta,\phi} \mathcal{L}_{\text{DDS}}), \tag{11}$$

where $\cdot$ denotes the pixel-wise multiplication and $\mathcal{M}$ is the conditional binary mask of the specific region of the target prompt to change. This mask is generated by utilizing off-the-shelf text prompt segmentation models such as CLIPSeg (Lüddecke & Ecker, 2022) and SAM (Kirillov et al., 2023) to segment the target region by a text prompt.

Despite the use of a binary mask, masked DDS loss $\nabla \mathcal{L}_{MDDS}$ update all parameters of NeRF potentially affecting even undesired areas. As a result, solely depending on the masked DDS loss may inadvertently result in alterations beyond the mask boundaries. Hence, we introduce an additional reconstruction loss as follows to mitigate undesired deformations beyond the mask.

$$\mathcal{L}_{\text{Mrec}} = \lambda_{im} \cdot \mathcal{M} \cdot \mathcal{L}_{\text{rtot}} + \lambda_{om} \cdot (1 - \mathcal{M}) \cdot \mathcal{L}_{\text{rtot}}. \tag{12}$$

Finally, the total editing loss is as follows:

$$\mathcal{L}_{\text{tot}} = \mathcal{L}_{\text{MDDS}} + \mathcal{L}_{\text{Mrec}} \tag{13}$$

By suppressing undesired alterations through the use of the masked reconstruction loss $\mathcal{L}_{Mrec}$, our total editing objective function updates NeRF and refinement layer $F_\theta$ and $F_\phi$, ensuring NeRF renders novel views in accordance with the desired text conditions.

## 4 EXPERIMENTAL RESULTS

### 4.1 BASELINE METHODS

To comprehensively evaluate the performance of our method, we perform comparative experiments comparing it to state-of-the-art methods. As CLIP-based text guidance editing methods, we used

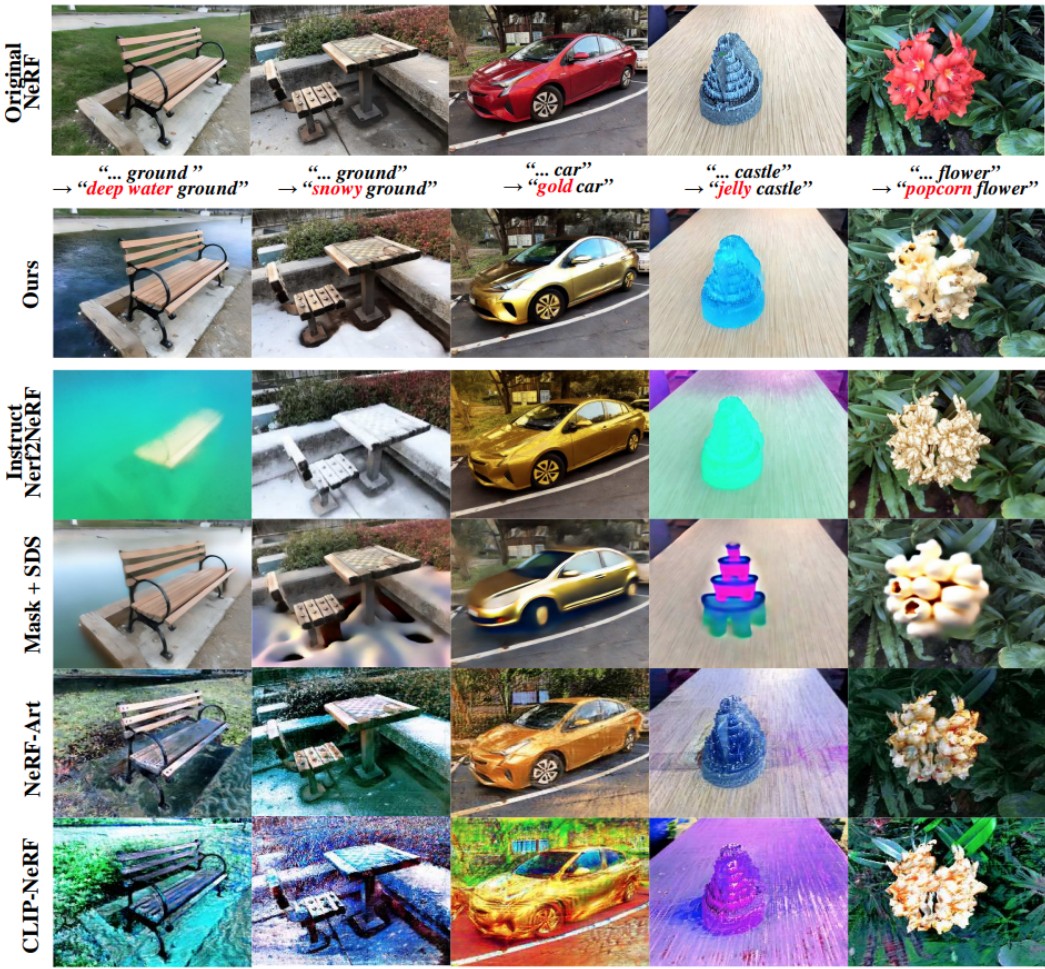

Figure 4: **Comparison with baseline models.** ED-NeRF demonstrates outstanding performance in effectively altering specific objects compared to other models. Baseline methods often failed to maintain the region beyond the target objects and failed to guide the model towards the target text.

CLIP-NeRF (Wang et al., 2022) and NeRF-ART (Wang et al., 2023a). CLIP-NeRF encodes the images rendered by NeRF to the CLIP embedding space, allowing it to transform the images according to the text condition. As an improved method, NeRF-ART trains NeRF with various regularization functions to ensure that CLIP-edited NeRF can preserve the structure of the original NeRF. For fair experiments, we re-implemented the methods to TensoRF backbone, referencing the official source codes. For the diffusion-based editing, we chose Masked SDS (Poole et al., 2022) and InstructNeRF2NeRF (Haque et al., 2023) as the methods that target local editing. In the masked SDS setting, we fine-tuned the pre-trained NeRF with applying basic SDS loss only to the masked regions so that the NeRF model is locally edited. InstructNeRF2NeRF (Haque et al., 2023) leverages the powerful generation capabilities of diffusion models to sequentially modify the entire dataset to align with text conditions and use the modified dataset as a new source for NeRF training. We utilized a database comprising real-world images, including LLFF (Mildenhall et al., 2019) and IBRNet (Wang et al., 2021) datasets, as well as the human face dataset employed in Instruction-NeRF2NeRF (Haque et al., 2023).

## 4.2 QUALITATIVE RESULTS

**Text-guided editing of 3D scenes.** As shown in Figure 1, our method shows its capability to edit various image types with different textual contexts. Specifically, it is possible to achieve the effective transformation of specific objects without affecting other parts. Our baseline method Instruct-NeRF2NeRF (Haque et al., 2023) shows decent results with high consistency between images and

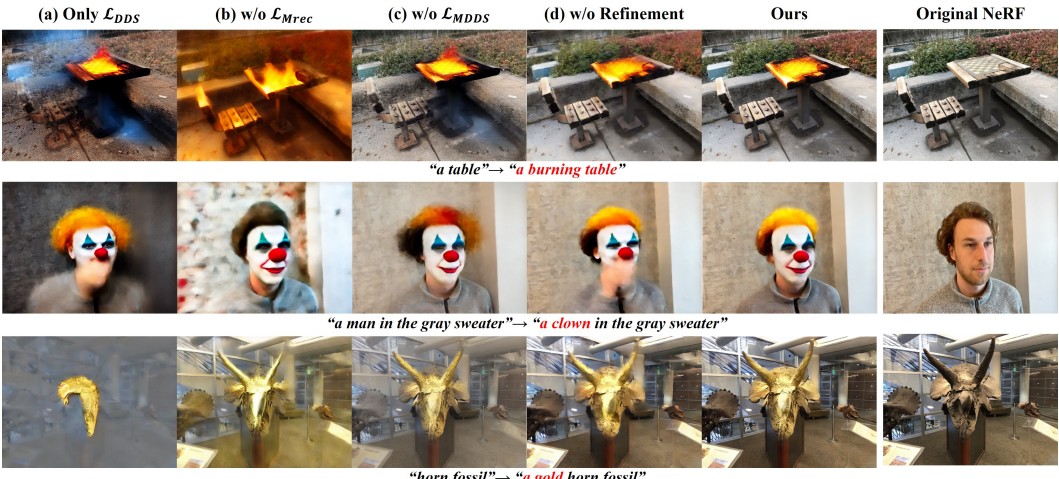

| (a) Only $\mathcal{L}_{DDS}$ | (b) w/o $\mathcal{L}_{Mrec}$ | (c) w/o $\mathcal{L}_{MDDS}$ | (d) w/o Refinement | Ours | Original NeRF |

*"a table"→ "a burning table"*

*"a man in the gray sweater"→ "a clown in the gray sweater"*

*"horn fossil"→ "a gold horn fossil"*

Figure 5: **Ablation studies.** (a) If we only use DDS loss, the model fails to maintain the attribute of untargeted regions and often fails to reflect text conditions. (b) If we do not use masked reconstruction regularization, again the regions beyond the target objects are excessively changed. (c) If we remove the mask from DDS, unwanted artifacts occur in untargeted regions. (d) With removing the proposed refinement layer, the results become blurry as the backbone NeRF cannot fully embed real-world scenes. Our proposed setting can modify a specific region in a 3D scene and follow the target word without causing unwanted deformations.

text conditions, as well as view consistency across scenes. However, it faces limitations in accurately transforming the specific objects to match text conditions and may introduce undesired image alterations beyond the specific objects. In Masked SDS, the edited output fails to reflect the structure of the original NeRF scene and shows unwanted artifacts. In the case of NeRF-ART, the entire image is embedded into the CLIP space, and it does not inherently recognize and modify only specific objects. Therefore, it exhibits limitations in recognizing and altering specific objects. CLIP-NeRF also encodes the images rendered by NeRF to the CLIP embedding space, allowing it to transform the images according to the text condition. However, its performance falls short when it comes to altering specific parts in a similar manner. On the other hand, our ED-NeRF exhibited powerful abilities in editing 3D scenes by specifying certain parts through text, surpassing other models. It not only excelled in changing objects but also demonstrated the capability to faithfully follow and modify areas that are not objects, such as the ground, in accordance with the text condition.

## 4.3 QUANTITATIVE RESULTS

**CLIP Directional Score.** In order to quantitatively measure the editing performance, we show the comparison results using CLIP Directional scores(Gal et al., 2021). The CLIP Directional score quantifies the alignment between textual caption modifications and corresponding image alterations. We rendered multiple view images from NeRF and measured the average score over images. When compared to baseline methods, our model obtained the best similarity scores. The result indicates that our edited NeRF accurately reflects the target text conditions.

**User Study.** In order to further measure the perceptual preference of human subjects, we conducted an additional user study. For the study, we rendered images from edited NeRF using 5 different scenes from LLFF and IBRnet. We gathered feedback from 20 subjects aged between their 20s and 40s. Each participant was presented with randomly selected multi-view renderings from our model and baselines and provided feedback through a preference scoring survey. We set the minimum score as 1 and the maximum score is 5, and users can choose the score among 5 options: 1-very low, 2-low, 3-middle, 4-high, 5-very high. To measure the performance of editing, we asked two questions for each sample: 1) Does the image reflect the target text condition?(Text score) 2) Does the model accurately edit the target object?(Preservation). 3) Does the 3D scenes preserve view consistency?(view consistency). In Table 1, we show the user study results. Compared with baseline methods, our method showed the best score in text score and preservation, and second best in view consistency. Overall, ours outperformed the baseline models in perceptual quality.

| Metrics | CLIP-NeRF | NeRF-Art | Instruct N2N | Mask SDS | Ours |
|---|---|---|---|---|---|
| CLIP Direction Score ↑ | 0.1648 | 0.1947 | 0.2053 | 0.1409 | **0.2265** |
| Text score↑ | 2.56 | 3.20 | 3.29 | 3.14 | **3.88** |
| Preservation ↑ | 2.30 | 2.97 | 3.08 | 2.76 | **4.09** |
| view consistency ↑ | 3.21 | **3.79** | 3.28 | 3.56 | 3.64 |

Table 1: **Quantitative Comparison.** We compared the text-image similarity between the target text and rendered output from edited NeRF (CLIP Directional Score). Also, we show the user study results in three categories text-guidance score, source preservation score, and view consistency. The results show that ours shows improved perceptual score among baseline models.

| Metrics | CLIP-NeRF* | NeRF-Art* | Instruct N2N | Ours |
|---|---|---|---|---|
| Fine-tuning time↓ | 6min | 15min | 90min | 14min |
| GPU Memory↓ | 17GB | 18GB | 15GB | 8GB |

Table 2: **Efficiency Comparison.** We compared the efficiency of ours and baseline methods in terms of training time and Memory usage. Our method can enable faster editing with lower memory usage. For CLIP-NeRF and NeRF-Art, the models are fine-tuned in lower resolution (252×189), due to excessive memory consumption. Instruct N2N and ours are fine-tuned in 512x512 resolution.

**Efficiency comparison.** To compare the editing efficiency, we check the fine-tuning time and memory usage in Table 2. Among baselines, our method uses the lowest memory for training, with a much lower time compared to Instruct Nerf2Nerf. GPU memory usage and training time are measured based on the RTX 3090. In the baselines of CLIP-Nerf and Nerf-art, we experiment with using downsized images as higher resolution editing causes GPU memory overflow. For Instruct Nerf2Nerf, the fine-tuning process requires excessive time as it periodically translates the training images. Considering that our method shows outperforming quality in text-guided editing, our proposed scheme is efficient in both memory and time aspects. When comparing the time for the pre-training NeRF backbone model, we did not include a comparison since all baselines and ours take almost the same amount of time (about 10 minutes). More Details and comparisons on pre-training time are in our Appendix.

## 4.4 Ablation Studies

To evaluate our proposed components, we conducted an ablations study in Figure 6. (a) If we only use DDS, the method fails to maintain the untargeted regions with artifacts, even failing in training (e.g. fossil). (b) If we do not use regularization $\mathcal{L}_{Mrec}$, the edited results show the target text attribute, but again the regions beyond the target objects are severely degraded. (c) When we remove mask guidance on DDS, (w/o $\mathcal{L}_{MDDS}$), unwanted minor deformations occur due to the gradient of DDS affecting the regions outside the mask. (d) When we remove our refinement layer, the results show blurry outputs, which indicate that latent NeRF is not accurately trained. When we utilize all the components we proposed, we can reliably transform the 3D scene into the desired target object while preserving the original structure source NeRF. In the Appendix, we included an ablation study on our proposed refinement layer for novel-view reconstruction tasks.

## 5 Conclusion

In this paper, we introduced a novel ED-NeRF method optimized in the latent space. By enabling NeRF to directly predict latent features, it efficiently harnesses the text-guided score function of latent diffusion models without the need for an encoder. By doing so, our approach is able to effectively reduce computation costs and address the burden of previous models that required rendering at full resolution to utilize the diffusion model. We extended the strong 2D image editing performance of DDS to the 3D scene and also introduced a new loss function based on the mask. As a result, it showed high performance in object-specific editing, a task that previous models struggled with. We experimented with our proposed approach across various datasets, and as a result, it demonstrated strong adherence to text prompts in diverse scenes without undesired deformation.

## 6 ETHICS AND REPRODUCIBILITY STATEMENTS

**Ethics statement.** ED-NeRF enables efficient and accurate text-guided NeRF editing, which can be applied to various applications. However, our ED-NeRF can be used for creating obscene objects which may cause the users to feel offended. In order to prevent the possible side effects, we can use a filtered diffusion model that does not contain malicious text conditions.

**Reproducibility statement.** We detailed our experimental process and parameter settings in our Appendix. We will upload our source code to an anonymous repository for reproduction.

## 7 ACKNOWLEDGEMENT

This research was supported by National Research foundation of Korea(NRF) (**RS-2023-00262527**), Field-oriented Technology Development Project for Customs Administration through National Research Foundation of Korea(NRF) funded by the Ministry of Science & ICT and Korea Customs Service(**NRF-2021M3I1A1097938**), the Korea Medical Device Development Fund grant funded by the Korea government (the Ministry of Science and ICT, the Ministry of Trade, Industry and Energy, the Ministry of Health & Welfare, the Ministry of Food and Drug Safety) (Project Number: 1711137899, KMDF_PR_20200901_0015) and Culture, Sports and Tourism R&D Program through the Korea Creative Content Agency grant funded by the Ministry of Culture, Sports and Tourism in 2023

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

## A    EXPERIMENTAL DETAILS

**Training ED-NeRF details.** For optimizing NeRF in Latent space, we use TensoRF as the backbone for fast and efficient rendering with up to 64 resolution by supervision. As stated in the original TensoRF, we use Adam optimizer. For RGB NeRF, the color values to predict are constrained within the range of 0 to 1. In the case of NeRF trained in the latent space, the range of latent values is at least between -3 and 3, which leads to faster convergence than density. Therefore, we set the learning rate for trainable density voxels, which form the sigma in the original TensoRF, to 0.04, while keeping the learning rates for other components the same as TensoRF at 0.02 during training. During the training process, ED-NeRF needed to form a feature map of size 64 resolution, therefore we configured the batch size to be 4096-pixel rays. We train ED-NeRF in latent space with source latent feature map supervision for 100k steps on the single NVIDIA RTX 3090. When fine-tuning ED-NeRF for editing purposes, we train ED-NeRF for 5k steps. We set $\lambda_{om}$ as 100 and $\lambda_{im}$ as 0.01 to strongly preserve the regions outside the mask while editing and reflecting the objects within the mask.

**Generate source image and latent feature map.** Datasets such as LLFF and IBRNet typically contain categories with a limited quantity of images. As mentioned earlier, due to the lack of guaranteed geometric consistency within the latent space, synthesizing views using NeRF with a limited dataset is an exceptionally challenging task. To address this issue from an editing perspective, we leveraged a pre-trained NeRF model trained on RGB images. We employed publicly available NeRF models for novel view synthesis, and when required, we trained our own RGB NeRF. Using this pre-trained NeRF, we generated novel view images by capturing over 100 views along a spiral camera path. Subsequently, these generated images were utilized as part of the source image dataset, denoted as $I$. The required number of iterations for convergence to the target scene can vary depending on the dataset, influenced by the provided images and text prompts.

## B    REFINEMENT LAYER DETAILS

**Architecture of Refinement layer.** We used an additional layer to correct the rendered latent. The front layer of the decoder embeds the 4-channel latent to a 512-channel vector before performing self-attention. The rear layer of the encoder calculates self-attention on a 512-channel vector and then samples a 4-channel latent through ResNet. Taking inspiration from this, in designing the refinement layer, we use the combined architecture of the decoder front layer and the encoder rear layer as shown in 2. We use 4 ResNet blocks and 2 self-attention blocks in our Refinement layer.

**Ablation study of Refinement layer.** To further show the effect of our proposed refinement layer, we show the comparison on different network architectures such as MLP and ResNet. For a fair comparison, both are designed to have the same depth as our Refinement layer. For the reconstruction results in Figure 6, we can observe that the reconstruction output from our refinement layer shows fine details when compared to the basic setting. As already shown in our previous ablation study in Figure 6, the refinement layer improves the latent space NeRF training which is crucial in the final edited output.

## C    EFFICIENCY COMPARISON DETAILS

To evaluate the effectiveness of our proposed ED-NeRF, we compared the overall fine-tuning time of text-guided editing. In the baseline methods, as the models apply editing on full-resolution NeRF backbones, they require a much longer time to obtain reasonable edited outputs. Especially for the Instruct-NeRF2NeRF, the overall editing time is over 30 minutes, up to 90 minutes based on text conditions, as there should be periodic replacement of edited 2D images. Therefore the training time is much longer than other baselines. In the case of CLIP-NeRF, the editing time is relatively short (about 6 min) compared to other baselines, but the overall quality is degraded as shown in our main script. Although NeRF art uses a similar training scheme to CLIP-NeRF, the model takes longer time over 15 minutes as the method uses additional regularizations for editing. In the case of NeRF-Art and CLIP-NeRF, we could not fine-tune the model with the original resolution due to memory overflow, therefore we downsampled the images to 252x189 resolution. In Masked SDS the average editing time is about 8 minutes, but again the output quality is not better than other

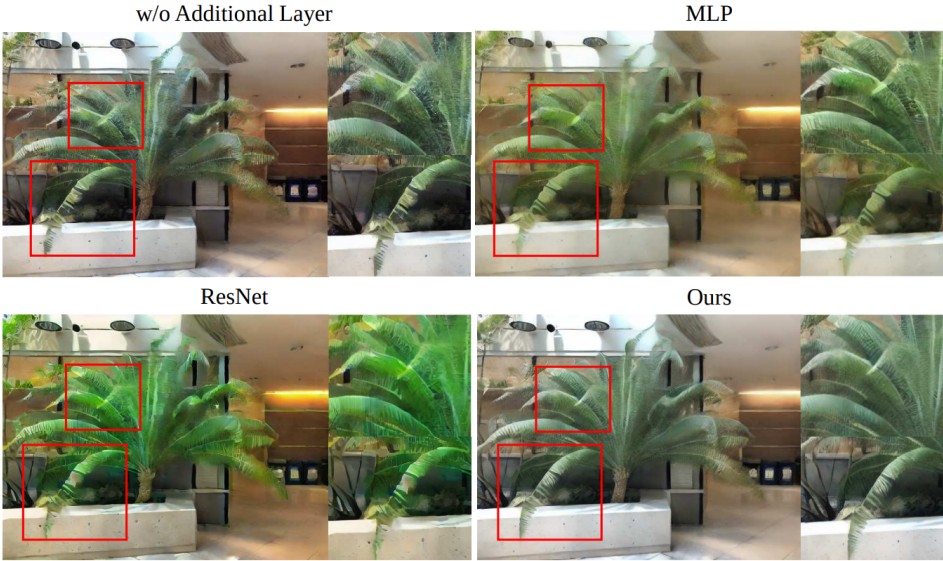

Figure 6: **Ablation study on refinement layer.** Novel view synthesis results with different architectures on additional refinement layers. Without refinement layer, MLP and ResNet make saturated or blurry images, but our refinement layer can represent high-frequency details.

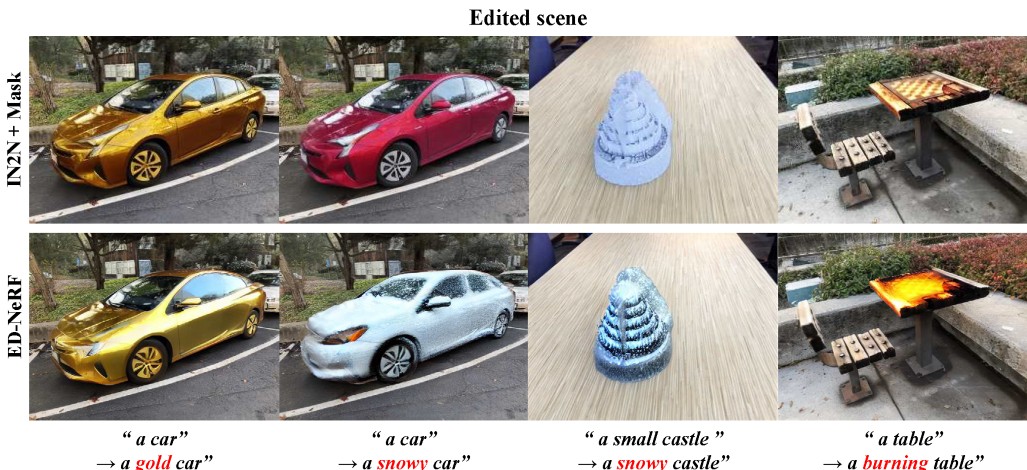

Figure 7: **Additional Comparison.** We show comparison results between our proposed ED-NeRF and mask-guided Instruct NeRF2NeRF. Instruct NeRF2NeRF often fails to capture the correct textual information.

NeRF Scene          Edited novel scene

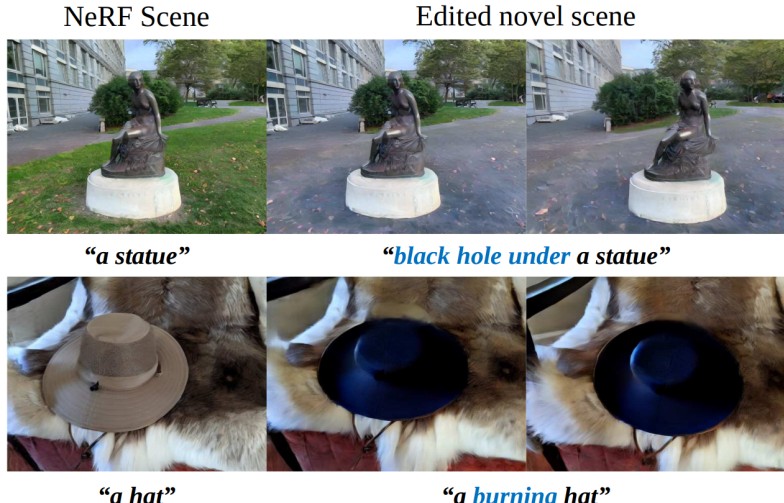

*"a statue"*          *"*black hole under* a statue"*

*"a hat"*          *"a* burning *hat"*

Figure 8: **Failure case.** The diffusion model is a 2D-based image generation model, so it may not consistently perform well in 3D image editing tasks. In this example, we aimed to change 'statue' to 'black hole under a statue' and 'a hat' to 'a burning hat.'

methods. In our case, when it comes to easy cases such as color change the fine-tuning time takes less than 3 minutes. In more complex cases, the training time is up to 14 minutes. In the case of CLIP-Nerf and mask-SDS, the longer training could not get improvement. Compared with other state-of-the-art baselines such as Instruct NeRF2NeRF and Nerf-Art, our method shows better time efficiency in training. Unlike other methods, ED-NeRF is able to achieve a shorter fine-tuning time because, for the synthesis of a 512x512 image, our method only needed to render a latent of size 64x64.

Furthermore, we compare the time it takes for the existing RGB NeRF model and ED-NeRF to generate 512x512 RGB images. Based on the LLFF dataset, TensoRF takes an average of 2.7143 seconds to render a single 512-resolution image. In contrast, our ED-NeRF, which renders the latent feature map and decodes it into an RGB image, takes an average of 1.1240 seconds per image. As a result of training in the latent space and reducing the resolution, ED-NeRF shortens the inference time by 2.4 times.

In terms of comparing the pre-training time, there is no absolute criteria for comparison since there are multiple choices of state-of-the-art NeRF backbones. In our baseline methods of CLIP-NeRF and NeRF-Art, we used the backbone model of TensoRF (Chen et al., 2022) which requires about 10 minutes for training. In the case of Instruct N2N, the method uses the Nerfacto (Zhang et al., 2021) model which also requires about 9 minutes for pretraining the original NeRF backbone. In our case, we need another step to extract novel view RGB images for training latent feature maps. For novel view extraction, we used the fastest public models such as Plenoxels (Fridovich-Keil et al., 2022), which takes less than 2 minutes for training and rendering. For encoding the RGB images to latent features, we can obtain the features using a single feedforward process with Stable Diffusion VAE encoder. This process takes less than 10 seconds. With the calculated latent features maps, we trained the Latent-embedded NeRF model which takes about 9 minutes with TensoRF framework. Therefore, the total pre-training time for our case is about 12 minutes. Overall, there is no large difference in pre-training time between ours and baseline methods. Considering there are other state-of-the-art NeRF backbones that show faster training, we can improve the time consumption for the pre-training stage in future work.

## D    ADDITIONAL COMPARISON RESULTS

**Comparison with IN2N incorporating an additional mask.** For further comparison, we conducted experiments between ours and mask-guided Instruct NeRF2NeRF (IN2N). As IN2N uses edited 2D images for training images, we can directly apply the binary mask on the training image

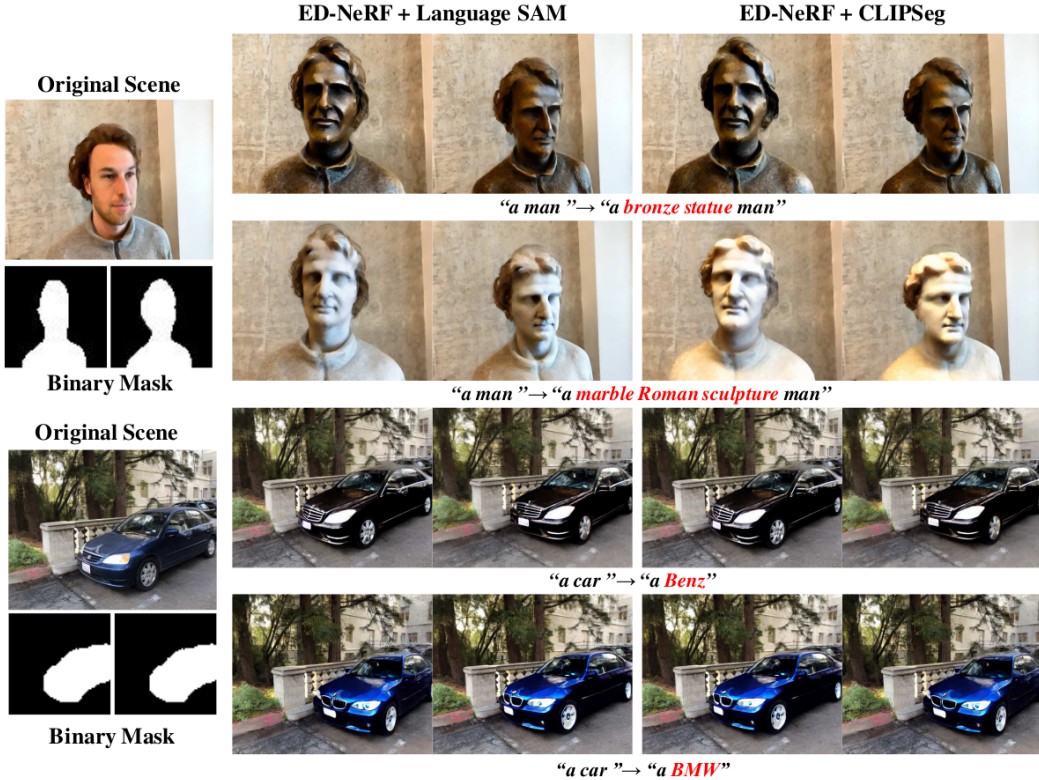

Figure 9: **Editing results using different segmentation models.** To evaluate the validation of our method, we experimented with editing using masks obtained from segmentation models with varying accuracies. We observed stable editing performance even when using different off-the-shelf models that allow the use of a text prompt as a condition.

so that only the target object area is edited. In Fig. 7, we show the comparison results. With the mask guidance, IN2N shows improved performance in local editing, as it does not change the unrelated regions. If the text prompt is relatively simple (e.g. gold car), the output quality of both models is high. However, as IN2N heavily relies on the performance of Instruct Pix2Pix (InP2P) model, the method cannot properly change the semantics of the objects if the editing prompt is difficult for InP2P (e.g. snowy car, burning table, etc.). The results show that our proposed method is more robust on text conditions when compared to baseline IN2N.

## E   BINARY MASK DETAILS

**Binary mask pipeline.** To create a binary mask, we consider an off-the-shelf segmentation model. Specifically, we consider models that can perform segmentation using text prompts. From well-performing models such as LanguageSAM (Kirillov et al., 2023; Medeiros, 2023) and CLIPSeg (Lüddecke & Ecker, 2022), we can extract binary masks from images. The mask generation process begins by using the segmentation model to obtain a binary mask from entire source images. At this point, a text prompt corresponding to the part of the image we want to modify is given as a condition, extracting a mask for the specific object. Since the mask needs to be applied alongside a 64×64-sized latent vector, it undergoes resizing to fit the dimensions of 64×64.

**The ablation study of segmentation models.** To explore the sensitivity of ED-NeRF in relation to the segmentation model, we utilized distinct binary masks extracted from different segmentation models. As illustrated in Figure 9, despite employing binary masks with differing performance from distinct mask models, we consistently obtain nearly identical NeRF editing results. In the figure, the left mask, derived through the LanguageSAM model, and the right mask, obtained through the CLIPSeg model, represent different masks extracted with 'man' and 'car' text prompts as conditions

**Linear decoded scene**

Figure 10: **Latent Space Visualization.** To verify the relationship between the RGB image and its corresponding latent feature map, we show the visualization outputs. Latents in the figure are linearly decoded results through matrix multiplication with a $3\times4$ matrix and a $4\times64\times64$ latent vector. For visualization purposes, we resize RGB images to $512\times512$ and upscale the $64\times64$ size decoded latent to $512\times512$ resolution. For all the cases, we can clearly see that both the images and latents have the same spatial information.

## F    LATENT FEATURE MAP ANALYSIS

In order to show whether the encoded latent representation actually has geometrical similarity to corresponding RGB images, we show the visualization results of latent in Fig.10. As our latent variable has a dimension of $4\times64\times64$, we cannot directly visualize them into an RGB image. Therefore, we used a linear decoding matrix that can calculate the corresponding RGB images from latent representations (Metzer et al., 2023; Turner, 2022). The results show strong evidence of a high correlation between RGB images and the corresponding latent representations.

## G    LIMITATIONS

Since we leverage a pre-trained Stable Diffusion model, there should be limitations in the generative performance of the diffusion model itself. For example, as shown in Figure 8 if we try to apply text conditions that are very far from the source object (e.g., 'a statue' to 'black hole under a statue'), sometimes the output quality is degraded. This may be solved when we use an improved version of the latent diffusion model.

## H    ADDITIONAL RESULTS

**Qualitative additional editing results.** To further show the edited results, we show multi-view rendered outputs from our edited NeRF in Fig. 13. The top scenes are altered from a reference to 'statue' to 'gold statue' and the scene just below it is changed from 'ground under the statue' to 'water pond ground under the statue.' In the case of the car images, 'SUV' is modified to 'gold SUV,' followed by 'shallow water pond under the SUV.' Lastly, in the picture of the man, it changes from 'a man in the gray sweater' to 'a man in the batman suit,' and subsequently to 'Einstein in the gray sweater.' Through our unique method, we were able to achieve a very high-quality edited 3D

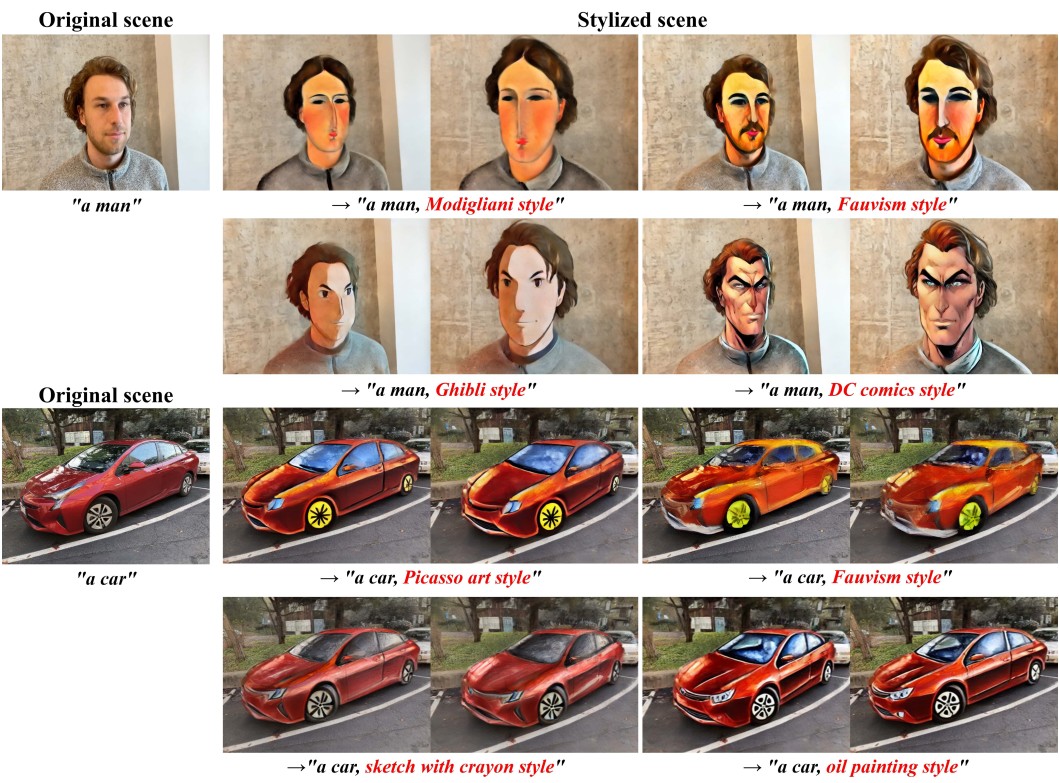

Figure 11: **Style transfer results.** By adding style text conditions to the target prompt, we can obtain high-quality stylized 3D scenes. The results show that the overall textures are changed to target text prompts while preserving the original scene structures.

| Ours | Ours + SDS | Ours | Ours + SDS |
|------|-----------|------|-----------|

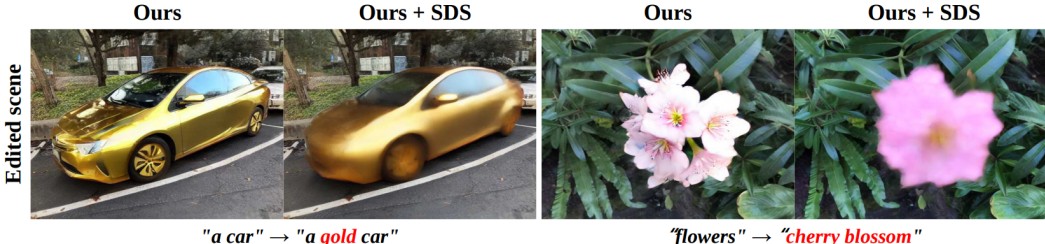

*"a car" → "a gold car"*    *"flowers" → "cherry blossom"*

Figure 12: **Editing results of using DDS with SDS** We additionally explore the editing of NeRF by incorporating DDS and SDS into our proposed method. As depicted in the figure, when using SDS to guide editing NeRF, the drawbacks of SDS continue to impact NeRF editing.

scene. In particular, our model could even depict objects in the original scene being reflected in the water when we changed the target region to a 'water pond'.

**NeRF Style transfer.** To verify the versatility of our proposed method, we experimented with 3D style transfer in Fig. 11. With the artistic style text prompts, our method can change the overall texture of objects while preserving the structure of original source images. Just adding a style condition text to the target prompt, we can obtain high-quality results as shown in the figure. (e.g. 'a car' → 'a car, Modigliani style') During the style base editing, we set $\lambda_{om}$ as 100 to preserve the overall structure of the edited object.

**Editing with DDS + SDS.** For further evaluation, we conduct experiments to examine the results of using DDS and SDS together in NeRF editing. Incorporating the target prompt-guided score obtained during the DDS process as an additional gradient, we used added gradient from DDS and SDS for fine-tuning. As shown in Figure 12, when DDS and SDS are combined, the image adheres to the text condition but undergoes a blurry transformation. This underscores the lingering issue with SDS mentioned earlier, deteriorating image editing outcomes.

**Visualized results of latent embedded NeRF in a natural scene.** In Figure 14, we present the rendering results of the ED-NeRF trained in the latent space. These results represent the state before changing the image with the target text, demonstrating that ED-NeRF can effectively represent real-world 3D scenes with a 64-resolution latent feature map and depth map. As depicted in Figure 2 (c), the RGB image is obtained by rendering the 64-resolution feature map with ED-NeRF, followed by decoding to generate a 512-resolution image, which is then resized to $504 \times 378$. The depth map, on the other hand, is directly resized from 64 resolution to $504 \times 378$.

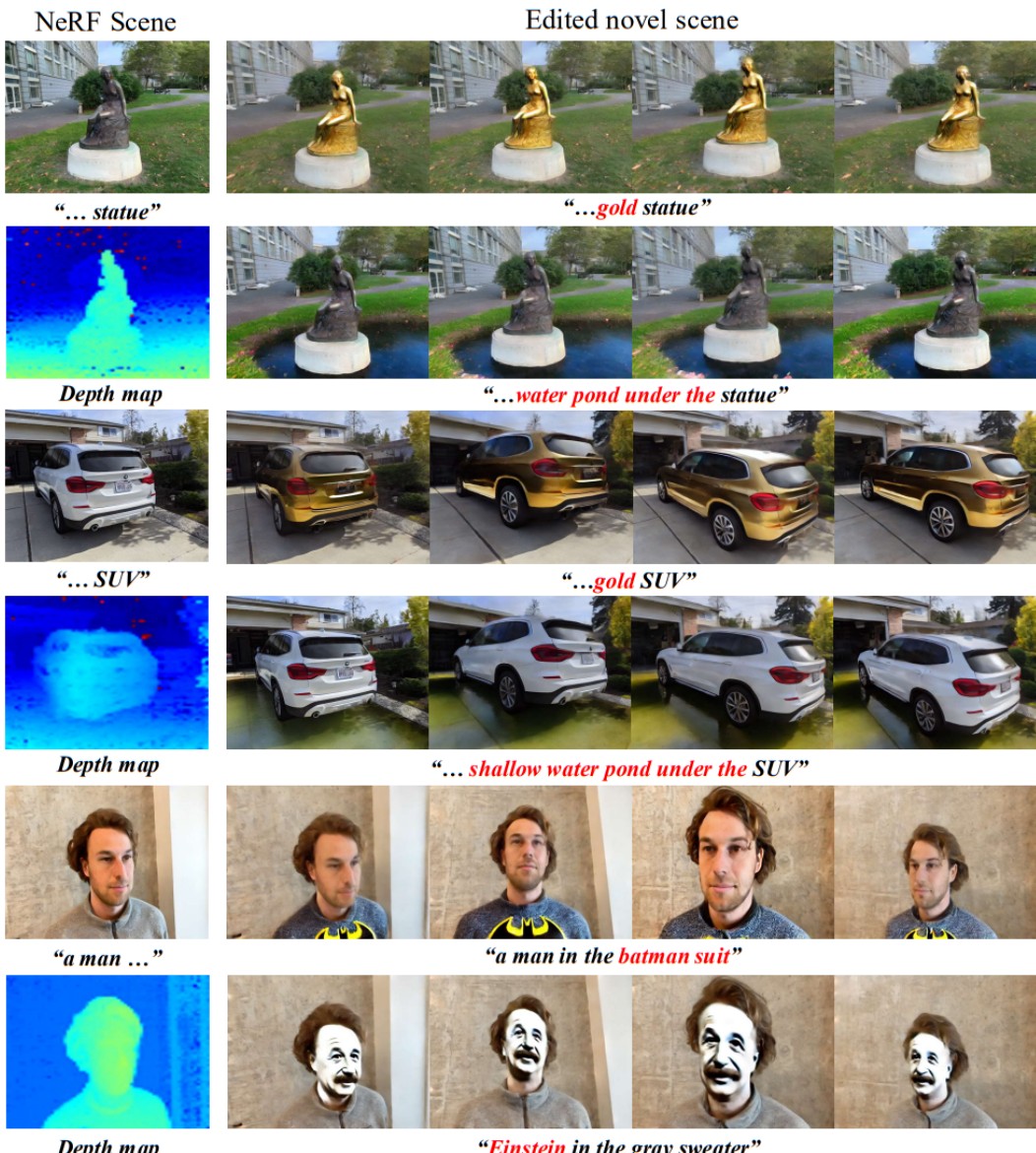

Figure 13: **Edited scene in various views.** We conduct an experiment where we manipulate different objects with text prompts in various views. It demonstrates the ability to edit various elements within 3D scenes and verify their accurate transformation in response to the provided text prompts. ED-NeRF has a depth map for the training scene in the latent space, with a resolution of 64. For visualization purposes, we resize it to 504×378 resolution in the above image.

ED-NeRF optimized in latent space of Stable Diffusion

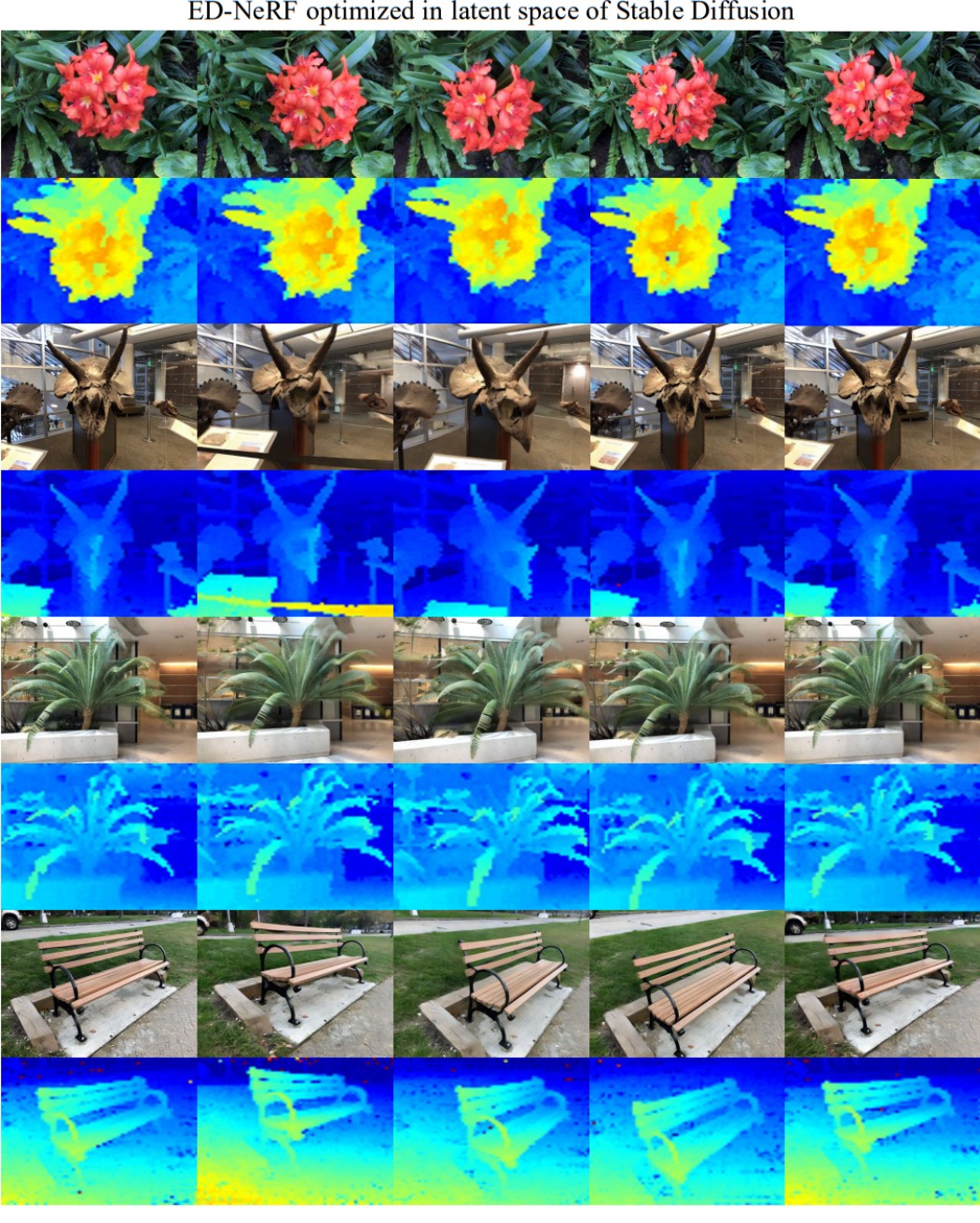

Figure 14: **Visualized latent feature and depth map.** As illustrated in the figure, the ED-NeRF trained in the latent space is capable of generating a high-quality depth map. We show visualizations of the features rendered by ED-NeRF from various scenes and angles before the editing phase. RGB images are the result of decoding latent feature maps rendered by our NeRF using Stable Diffusion, and they are resized from $512 \times 512$ resolution to $504 \times 378$ resolution for visualization purposes. Similarly, the depth maps are formed alongside the latent feature map at a resolution of $64 \times 64$, and it has also been resized to $504 \times 378$ resolution for visualization.

