# OpenReview forum: "ED-NeRF: Efficient Text-Guided Editing of 3D Scene With Latent Space NeRF"
_ICLR.cc/2024/Conference — ICLR 2024 poster_

### Official Review · Reviewer_kJsR · 2023-10-30

**Soundness:** 3 good
**Presentation:** 3 good
**Contribution:** 2 fair
**Rating:** 6
**Confidence:** 3

**Summary:**

This paper proposes a method for NeRF editing by incorporating the latent diffusion model and masked segmentation to control the editing of rendered views. Unique in the process is the iterative refinement process which ensure the multi-view consistency in NeRF, while ensuring the editing agrees with the target latent direction. The proposed method is compared with InstructNeRF2NeRF and other NeRF editing methods which show superior results over the compared methods.

**Strengths:**

The proposed method is straight forward and it demonstrates good editing results on large variety of examples.

**Weaknesses:**

After checking the paper in detail, I got an impression that the improved performance are mainly due to the usage of object mask, instead of the DDS proposed in the paper. This impression is strengthen after checking the experimental comparisons with the InstructNeRF2NeRF. In that sense, I also feel that the comparisons are entirely fair since the compared method does not use any object mask. The paper also did not mentioned how these masks are obtained and how sensitive is the proposed method against the object mask segmentation accuracy. Overall, I think this is a good attempt for NeRF editing, but I somewhat feel that this is an incomplete submission if the authors cannot resolve my concerns above.

**Questions:**

Please correct me if I have ever made mistakes in my evaluations.

---

> ### Author Response · Authors · 2023-11-18
> **Reply to Reviewer kJsR**
>
> > **Q1 :After checking the paper in detail, I got the impression that the improved performance are mainly due to the usage of object mask, instead of the DDS proposed in the paper. This impression is strengthen after checking the experimental comparisons with the InstructNeRF2NeRF. In that sense, I also feel that the comparisons are entirely fair since the compared method does not use any object mask.**
>
> - To address the reviewer's concern, we included an additional comparison with masked Instruct-NeRF2NeRF in the Appendix of our revised manuscript. When we apply a regional mask on In2n, we can improve the local editability without changing the unwanted regions. However, the output quality is still not much improved showing that the output often fails to follow the text condition. We conjecture that even though we use masked guidance, Instruct N2N performance is largely affected by the editing performance of Instruct Pix2Pix. As In2n directly uses the edited 2D outputs as source images, the translation performance of Instruct Pix2Pix can directly affect the output quality. Therefore, the output quality can be degraded if the image sets are not properly changed from InP2P.
>
> > **Q2: The paper also did not mentioned how these masks are obtained and how sensitive is the proposed method against the object mask segmentation accuracy.**
>
> - Thank you for your constructive comment. Per the reviewer's request, we conducted another ablation study on different mask-generation methods. Please see the Appendix in the revised paper. Also, we elaborated on the process for mask generation in our appendix.

---

> ### Author Response · Authors · 2023-11-21
> **Nearing the end of the discussion period**
>
> Dear reviewer kJsR,
>
> As the deadline for the reviewer-author discussion phase is fast approaching (there is only a day left), we respectfully ask whether we have addressed your questions and concerns adequately. We would be happy to clear up any additional questions.

---

> ### Author Response · Authors · 2023-11-23
> **[Reminder] Summarization of our rebuttal**
>
> Dear reviewer kJsR,
>
> We believe that we have addressed the concerns that you have raised. Specifically,
>
> 1. We have added **experimental comparisons** with **masked InstructNeRF2NeRF** in the revised appendix.
>
> 2. We included an explanation of **Mask processing detail**
>
> 3. We have also added **comparative experiments** utilizing **segmentation models with different accuracies** to the appendix.
>
> We would like to gently remind you that the end of the discussion period is imminent. We would appreciate it if you could let us know whether our comments addressed your concerns.
>
> Best regards,
> Authors

---

### Official Review · Reviewer_mMij · 2023-10-30

**Soundness:** 4 excellent
**Presentation:** 2 fair
**Contribution:** 3 good
**Rating:** 6
**Confidence:** 3

**Summary:**

The paper proposes an improved 3D NeRF editing pipeline to overcome existing challenges like long training time, instability, and dropped image quality. The key idea is to bring NeRF training into the latent space of Stable Diffusion (SD), and perform all editing on the rendered latent feature directly. A refinement layer is introduced to enable nearby pixel feature interactions thus enhancing rendering quality, while an adapted mask-aware DDS loss (compared to the common SDS loss) is utilized to distill the SD guidance more effectively and ease the color saturation and mode-seeking. Various ablations analysis, comparison to baselines have demonstrated its effectiveness and superior performance against previous methods.

**Strengths:**

[**Significance**]
- Inspired by Latent-NeRF, the author proposes to train and edit NeRF in VAE latent space, which can help reduce time and computation costs, this is a smart and important move for NeRF editing related works.
- Their results have surpassed existing methods both quantitatively and visually;

[**Originality**]
- The author further proposes the refinement layer to let pixel correlations help increase model capacity, which overcomes the major drawbacks of quality drifting when using ray-based rendered latent feature for NVS. The proposed modification is simple yet effective.
- While other works tend to follow existing SDS-based distillation scheme, the author is well aware of the limitations of SDS loss, and further proposes to borrow recent advancements in image editing to help 3D editing. Though DDS method is not developed by the authors, the application and extension in 3D is novel;

[**Clarity** and **Quality**]
I like the figure 2 and figure 3 for pipeline design, which is clear and easy to understand. The author has compared with various baselines, and conducted a lot of visual comparisons, which is informative, and clarified well.

**Weaknesses:**

- The paper seems to be casual on writing, and there might be mistakes. Like Eq. 12 and Eq.13, why are L_{tot}  and L_{Mrec} part of each other? Further, there should be a formal table in the main paper on computing time comparison, rather than coarse text in the supp. The Efficiency improvements are one of the major claim and contributions of the paper, it is even added into the title of the paper ": EFFICIENT TEXT-GUIDED...", thus putting the time comparison in supp. is not good.
- Lacking of object removal and insertion experiments. Maybe I'm wrong, but most examples shown in the paper seem to only about modifying the centric object material, identity, or the background, it doesn't show how the methods works about removing an object, or inserting another object.

**Questions:**

- It seems the view consistency is not as good as several previous methods, while the small gap is acceptable, I am wondering what would be an example of view inconsistency, this can help us understand the space to improve;
- In Eq. 9, I guess the order of these two items are wrong? The target SDS item should sit first, or it doesn't matter? I am also wondering what if we combine the original SDS loss in addition to the DDS loss, would it further improve the performance?
- Reflection is an issue in NERF if not handling well, and the tensor-RF backbone doesn't model that. However, it seem that Figure 4 the car example demonstrated the vivid reflection, is this true? If this is the case, should we give the credit to the refinement layer?

---

> ### Author Response · Authors · 2023-11-18
> **Reply to Reviewer mMij**
>
> > **Q1: The paper seems to be casual on writing, and there might be mistakes. Like Eq. 12 and Eq.13, why are L\_{tot} and L\_{Mrec} part of each other?**
>
> - Sorry for the typo and thank you for your comment. It’s a mistake in Eq. 12 and Eq 13 that L{tot} and L{Mrec} are part of each other. we changed these in the revised paper.
>
> > **Q2: Further, there should be a formal table in the main paper on computing time comparison, rather than coarse text in the supp. The Efficiency improvements are one of the major claims and contributions of the paper, it is even added into the title of the paper ": EFFICIENT TEXT-GUIDED...", thus putting the time comparison in supp. is not good**.
>
> - Thank you for your feedback. We included the efficiency comparison part in our main quantitative results section.
>
> > **Q3: Lacking of object removal and insertion experiments. Maybe I'm wrong, but most examples shown in the paper seem to only about modifying the centric object material, identity, or the background, it doesn't show how the methods works about removing an object, or inserting another object.**  **.**
>
> - Thank you for your comment. We kindly remind the reviewer that object removal or inpainting on 3D scenes is not within our task scope, and baseline methods also did not show the results on those tasks. Instead, we show the object replacement task in our project page (e.g. castle-\> croissant). That said, the suggested direction is important, which we will investigate as a follow-up work.
>
> > **Q4: It seems the view consistency is not as good as several previous methods, while the small gap is acceptable, I am wondering what would be an example of view inconsistency, this can help us understand the space to improve;**
>
> - To address the reviewer's concern, we show the Freeview rendering video on our project page [https://ed-nerf.github.io](https://ed-nerf.github.io). Our method shows view-consistent edited outputs. Although there are slight artifacts on the rendered outputs, we are improving the problems with more parameter tuning.
>
>
> > **Q5: In Eq. 9, I guess the order of these two items are wrong? The target SDS item should sit first, or it doesn''t matter? I am also wondering what if we combine the original SDS loss in addition to the DDS loss, would it further improve the performance?**
>
> - Sorry for the typo. We corrected them in our revised paper. Also, per the reviewer's request, we trained the model with a combination of DDS and SDS. Please see the results in our appendix. The results show that a combination of two losses does not bring an advantage, even making blurry outputs.
>
> > **Q6: Reflection is an issue in NERF if not handling well, and the tensor-RF backbone doesn't model that. However, it seem that Figure 4 the car example demonstrated the vivid reflection, is this true? If this is the case, should we give the credit to the refinement layer?**
>
> - Thank you for your interesting comment. We conjecture that expression of reflection comes from both of latent space NeRF with DDS and refinement layer. NeRF represents the color of a point through volume rendering, making it unable to capture surface reflections caused by material properties.[1] However, in the case of NeRF trained in latent space, the final output image is obtained through the decoder of Stable Diffusion from the rendered latent. This allows for the utilization of a rich representation that volume rendering alone cannot express. Additionally, when updating NeRF using DDS, considering the material properties of the target text prompt allows for the expression of rich reflections.(e.g. "gold") Moreover, by employing a refinement layer, our work effectively reduces blurry artifacts, enabling the representation of high-frequency details.
> [1]: NeRFReN: Neural Radiance Fields with Reflections}

---

> ### Author Response · Authors · 2023-11-21
> **Nearing the end of the discussion period**
>
> Dear reviewer mMij,
>
> As the deadline for the reviewer-author discussion phase is fast approaching (there is only a day left), we respectfully ask whether we have addressed your questions and concerns adequately. We would be happy to clear up any additional questions.

---

> > ### Comment · Reviewer_mMij · 2023-11-23
> >
> > Thanks for the authors's clear answer, my concerns have been well addressed.

---

> ### Author Response · Authors · 2023-11-23
> **Official Comment by Authors**
>
> Thank you for the discussion and the positive score!

---

### Official Review · Reviewer_rsnk · 2023-11-01

**Soundness:** 2 fair
**Presentation:** 3 good
**Contribution:** 2 fair
**Rating:** 5
**Confidence:** 5

**Summary:**

This paper presents a novel 3D NeRF-based approach for editing real-world scenes using a latent diffusion model (LDM) and a refinement layer. The proposed approach is faster and more amenable to editing than traditional NeRF methods. The authors conduct an analysis of the latent generation process and the proposed refinement layer based on that can significantly enhance performance. To further improve editing performance, the authors propose an improved loss function that is tailored for editing tasks by incorporating the delta denoising score (DDS) distillation loss. The experimental results show that the proposed approach achieves faster editing speed and improved output quality compared to traditional NeRF methods. Notably, the proposed approach effectively alters specific objects while baseline methods often fail to maintain the region beyond the target objects and fail to guide the model towards the target text.

**Strengths:**

1. The paper provides a well-motivated solution to the problem of editing pre-trained 3D implicit networks. The proposed approach effectively preserves the integrity of the original 3D scene while enabling desired modifications. This is a significant advantage as it ensures that the edited results are coherent and consistent with the original scene.

2. The incorporation of DDS into 3D for ED-NeRF editing is a reasonable and innovative adaptation. This allows for more precise editing of implicit 3D models, thereby enhancing the overall quality of the edited results.

3. The qualitative results presented in the paper demonstrate that the proposed technique can efficiently modify specific objects while preserving regions beyond the target objects. This is an improvement over baseline methods that often face difficulties in guiding the model toward the intended text. The ability to accurately edit specific objects while maintaining the integrity of the surrounding environment is a crucial advantage of the proposed method.

4. The author's provision of the necessary code enhances reproducibility.

**Weaknesses:**

1. Although the paper showcases impressive qualitative editing results, it does not provide free-viewpoint rendering results in video format. Consequently, it is hard to determine whether the proposed method can generate view-consistent editing. This is my primary concern regarding the acceptance of this submission, and I am open to increasing my rating if the author can provide video results.

2. The author presents a time analysis in the appendix; however, it is not comprehensive and does not sufficiently support the claim of "improved efficiency." The analysis only reports the time for inference, excluding training time. Given that the "Generate source image and latent feature map" paragraph in Appendix A states their approach involves generating novel view images and extracting their latent feature maps, the time required for this procedure should also be reported and factored in when comparing with baseline approaches.

**Questions:**

1. The paper presents an innovative approach for editing pre-trained 3D implicit networks while preserving the surrounding environment. It would be helpful if the author could provide additional insights into why their method can edit the targeted object while preserving other parts without introducing unwanted artifacts. This would enhance the understanding of the proposed method and its underlying mechanisms.

2. The paper mentions adopting a 4-channel latent representation for the proposed method. It would be beneficial if the author could provide more details about the adopted latent feature vectors. Specifically, it would be helpful to know if this representation is strong enough for the task of editing, and how the feature maps are visualized since they appear visually similar to original RGB images.

3. The rendered depth map presented in the paper appears somewhat coarse. It would be useful if the author could explain this observation.

---

> ### Author Response · Authors · 2023-11-18
> **Reply to Reviewer rsnk (1/2)**
>
> > **Q1 : Although the paper showcases impressive qualitative editing results, it does not provide free-viewpoint rendering results in video format. Consequently, it is hard to determine whether the proposed method can generate view-consistent editing. This is my primary concern regarding the acceptance of this submission, and I am open to increasing my rating if the author can provide video results.**
>
> - Thank you for your constructive comment. To address the reviewer's concern, we made an anonymous project page that includes free-view rendered videos.See [https://ed-nerf.github.io].
>
> > **Q2 : The author presents a time analysis in the appendix; however, it is not comprehensive and does not sufficiently support the claim of "improved efficiency." The analysis only reports the time for inference, excluding training time. Given that the "Generate source image and latent feature map" paragraph in Appendix A states their approach involves generating novel view images and extracting their latent feature maps, the time required for this procedure should also be reported and factored in when comparing with baseline approaches.**
>
> - Thank you for your constructive comments. To address the reviewer's concern, we included a comparison of full processes including pretraining NeRF backbones in the appendix and quantitative results in the main script. Also, we included a comparison of GPU memory usage to further show the efficiency of our method.
> - Please See the details in the general comment and revised paper.
>
>
> > **Q3 : The paper presents an innovative approach for editing pre-trained 3D implicit networks while preserving the surrounding environment. It would be helpful if the author could provide additional insights into why their method can edit the targeted object while preserving other parts without introducing unwanted artifacts. This would enhance the understanding of the proposed method and its underlying mechanisms.**
>
> -  Taking the analysis from the original paper of DDS, we can consider the SDS gradient as ‘direction’ for guidance. In the paper, SDS gradient for the output z and target text y can be decomposed into two components $\nabla_\theta \mathcal{L}\_ {SDS} (z,y,\epsilon,t):=\delta\_{\text{text}}+\delta\_{\text{bias}}$. Here $\delta\_{\text{text}}$ is component which can properly guide the output to the target text direction, and $\delta\_{\text{bias}}$ is an interference term which makes the blurry and degraded outputs on unrelated regions. For the source image, we can also calculate SDS gradient $\nabla_\theta \mathcal{L}\_{SDS}(\hat{z},\hat{y},\epsilon,t)$. In this case, as our source latent and source text is aligned, $\delta\_{\text{text}}$ in this case should be zero in the optimal case. Therefore, we can assume $\nabla\_\theta \mathcal{L}\_ {SDS}(\hat{z},\hat{y},\epsilon,t) \approx \hat{\delta}\_{\text{bias}}$. As we add the same noise to both of target and source cases, we can also assume $\delta\_{\text{text}} \approx \hat{\delta}\_{\text{text}}$ which makes DDS gradient $\nabla_\theta \mathcal{L}\_ {SDS}(z,y,\epsilon,t)-\nabla\_\theta \mathcal{L}\_{SDS}(\hat{z},\hat{y},\epsilon,t) \approx \delta\_{\text{text}}$
> which is an accurate editing direction without interference from unrelated regions. Therefore DDS gradients can edit the local regions that are related to text condition by updating pixel parameters directly.[1] However, applying the DDS to parameter space, not pixel space, can make artifacts in the unwanted area because the gradient of DDS can affect the whole parameter update process. To address this issue, we preserve DDS updates outside of the target region and regulate those areas with a reconstruction loss. By reconstructing the areas outside the target region using the original latent representation during the editing process, our method proves effective as the rendered latent itself closely follows the text condition.
>
>       [1] Delta Denoising Score, A Hertz, K Aberman, et al., ICCV 2023
> - Upon the edit-friendly nature of DDS gradient, we can edit the 3D objects by fine-tuning NeRF models. However, we empirically found that vanilla usage of DDS still shows artifacts in unwanted regions, therefore we included proper regularization strategies for further improving the local editing performance.

---

> ### Author Response · Authors · 2023-11-18
> **Reply to Reviewer rsnk (2/2)**
>
> > **Q4 :The paper mentions adopting a 4-channel latent representation for the proposed method. It would be beneficial if the author could provide more details about the adopted latent feature vectors. Specifically, it would be helpful to know if this representation is strong enough for the task of editing, and how the feature maps are visualized since they appear visually similar to original RGB images.**
>
> - Thank you for your comment. We kindly remind the reviewer that there are several advantages when using latent representations **.**
> - Our latent representation has decreased dimension of 64x64x4, therefore we can reduce the GPU memory during training and also obtain faster rendering. As previous methods using models directly trained on RGB space require full-size image rendering for using them as input for Stable Diffusion, it requires large memory consumption for training. To further show the efficiency of our method, we show the comparison on time and memory aspect in the main script.
> - Please See the details in the general comment and revised paper.
> - To show whether the latent space actually contains the context of corresponding RGB images, we show the latent visualization using the linear decoding used in Latent-NeRF in our appendix. We can see the latent features and images have similar contexts and common spatial information.
>
> > **Q5: The rendered depth map presented in the paper appears somewhat coarse. It would be useful if the author could explain this observation.**
>
> - We kindly remind the reviewer that our Latent space NeRF is trained on dimension-reduced latent space Stable Diffusion. Therefore, our depth map extracted from the NeRF model also has a smaller depth map size of 64x64. Although the depth map resolution is reduced, we can see that the map contains geometry and distance information of original images, and the rendered output quality is still high. This indicates that our latent space NeRF has enough capacity to contain real-scene representations.

---

> ### Author Response · Authors · 2023-11-21
> **Nearing the end of the discussion period**
>
> Dear reviewer rsnk,
>
> As the deadline for the reviewer-author discussion phase is fast approaching (there is only a day left), we respectfully ask whether we have addressed your questions and concerns adequately. We would be happy to clear up any additional questions.

---

> ### Author Response · Authors · 2023-11-23
> **[Reminder] Summarization of our rebuttal**
>
> Dear reviewer rsnk,
>
> We believe that we have addressed the concerns that you have raised. Specifically,
>
> 1. In order to demonstrate view-consistent editing results, we have uploaded the **free-view rendering videos** to the **project page** [https://ed-nerf.github.io/].
>
> 2. To assess improved efficiency, we have added a **comparative table** measuring **training time** and **GPU memory consumption** in the revised paper.
>
> 3. We have explained why our method is effective in editing the target region.
>
> 4. We have added an explanation of the **relationship** between **Latent and RGB maps** in the revised Appendix.
>
> 5. We have provided an explanation for why the **rendered depth map is coarse** and **its meaning.**
>
> We would like to gently remind you that the end of the discussion period is imminent. We would appreciate it if you could let us know whether our comments addressed your concerns.
>
> Best regards,
> Authors

---

### Official Review · Reviewer_EEVN · 2023-11-07

**Soundness:** 3 good
**Presentation:** 3 good
**Contribution:** 2 fair
**Rating:** 5
**Confidence:** 4

**Summary:**

This paper presents a method for text-guided 3D NeRF editing. Compared to the existing method, to solve the efficiency, it embeds real-world scenes into the latent space of the latent diffusion model (LDM) through a refinement layer. Moreover, it presents an improved loss function tailored for editing by exploiting (DDS) distillation loss and binary mask for accurate editing. The experiment validates the effectiveness of the proposed method.

**Strengths:**

1. The paper is in general well organized and easy to follow.
2. The DDS distillation loss with the binary mask is shown to be effective.

**Weaknesses:**

1. The key contribution of the proposed method lies in the integration of several small techniques, e.g., latent NeRF representation and DDS distillation loss, without too much novel insight. I am not pretty sure whether it is enough for the ICLR.
2. One of the motivations of the method is the efficiency of NeRF editing.  To this end, the latent NeRF representation is adopted, while the efficiency comparison with the existing methods is not shown in the main paper. Moreover, there is a lot of work for fast NeRF training/finetuning, which makes the improvement less interesting.
3. The design of Refinement layers is straightforward to me, and it is more like adding some additional layers after the volume rendering, which is a normal way to improve the rendering quality. Moreover, from Fig. 2, it looks like some layers are adopted directly from the SD VAE. What's the benefit of such a desgin?
4. In the experiment, most of the examples are about color changes or local texture editing. I am wondering whether the proposed method works for other 2D editing, such as the whole image style transfer or object replacement as shown in Instruct-NeRF2NeRF. Moreover, from Fig.4, we can see that compared to Instruct-NeRF2NeRF, the improvement lies in the accuracy of the editing for certain areas. I am wondering whether Instruct-NeRF2NeRF will perform better just with the mask constraint.

**Questions:**

Please refer to the weakness. Besides, there are two small questions.
1. It looks like that there is a typo in Fig. 2(c).
2. There are some minor typos in the paper.

---

> ### Author Response · Authors · 2023-11-18
> **Reply to Reviewer EEVN**
>
> >  **Q1 : The key contribution lies in the integration of several small techniques, e.g., latent NeRF representation and DDS distillation loss, without too much novel insight. I am not pretty sure whether it is enough for the ICLR.**
> - We kindly remind the reviewer that although previous work of Latent-NeRF [1] first proposed to train NeRF in the latent space of Latent Diffusion Models, the task of Latent-NeRF is only targeting to generate **'virtual'** 3D object from given text conditions, not to **'real'** 3D data. In contrast, our approach pioneers training Latent space NeRF for real 3D scenes by embedding real-scene RGB images as latent representations, improving reconstruction performance with a novel refinement layer. Training NeRF in latent space offers advantages such as reduced training burden and enhanced editability through direct use of rendered output in LDM models.
> - Therefore, we believe our work's contribution is significant as it achieves technical breakthroughs by showcasing the potential for expanding existing NeRF research, which predominantly concentrates on training with natural RGB.
> - In addition, our method is the pioneer in extending the DDS framework to the 3D editing space of NeRF, which was previously exclusive to 2D images. Unlike previous text-guided editing NeRF methods primarily relying on default SDS or CLIP models, our approach recognizes the unique editing capability of DDS in 3D NeRF editing. This novelty is evident in its significantly superior performance compared to using SDS. Our successful application of DDS to 3D NeRF editing marks a groundbreaking contribution.
>
>      [1] Latent-NeRF for Shape-Guided Generation of 3D Shapes and Textures, Metzer, Gal, et al., CVPR 2023
>
> > **Q2 : One of the motivations of the method is the efficiency of NeRF editing. To this end, the latent NeRF representation is adopted, while the efficiency comparison with the existing methods is not shown in the main paper. Moreover, there is a lot of work for fast NeRF training/finetuning, which makes the improvement less interesting.**
> - As reviewer commented, there are several off-the-shelf NeRF backbone models which enable faster training. However, editing a NeRF model with Diffusion model demands rendering high-resolution images (e.g., 512x512) and computing gradients with respect to it, leading to significant GPU memory consumption. In contrast, our model is trained on smaller-dimension data (e.g., 64x64), reducing memory requirements for fine-tuning the NeRF. By addressing the fundamental issue of training efficiency through data dimension reduction, we achieve more resource-efficient training compared to using NeRF models in image space. Efficiency comparisons are detailed in the general reply and revised paper.
>
> > **Q3 : The design of Refinement layers is straightforward to me, which is a normal way to improve the rendering quality. Moreover, from Fig. 2, it looks like some layers are adopted directly from the SD VAE. What's the benefit of such a design?**
>
> - We kindly remind the reviewer that we used a refinement layer as a bridge between the latent space of stable diffusion (SD) and neural radiance fields (NeRF) for the first time, making it different from conventional rendering improvement methods. From this perspective, we adopted the structure of an SD VAE with attention, which allows encoding that considers pixel correlation in the latent space. Additional experiments on various refinement layer structures, presented in the revised manuscript's appendix, validate that our proposed network structure exhibits the best reconstruction performance.
>
> > **Q4 : In the experiment, most of the examples are about color changes or local texture editing. I am wondering whether the proposed method works for other 2D editing, such as the whole image style transfer or object replacement as shown in Instruct-NeRF2NeRF. Moreover, from Fig.4, we can see that compared to Instruct-NeRF2NeRF, the improvement lies in the accuracy of the editing for certain areas. I am wondering whether Instruct-NeRF2NeRF will perform better just with the mask constraint.**
>
> - We conducted additional experiments on style transfer and object replacement. We also show the object replacement results on our project page (e.g. castle -\> croissant)(https://ed-nerf.github.io).
> We show the style transfer results in our revised Appendix.
> - To address the reviewer's concern, we added a comparison with masked Instruct-N2N in the revised Appendix. Applying regional masks to In2n enhances local editability without altering undesired regions, but the output quality often falls short of meeting text conditions. We conjecture that Instruct N2N performance is significantly influenced by Instruct Pix2Pix's editing capabilities. Since In2n utilizes the edited 2D outputs for training, the translation performance of Instruct Pix2Pix directly impacts output quality. Improper changes to image sets in InP2P can degrade the output quality.

---

> > ### Author Response · Authors · 2023-11-21
> > **Nearing the end of the discussion period**
> >
> > Dear reviewer EEVN,
> >
> > As the deadline for the reviewer-author discussion phase is fast approaching (there is only a day left), we respectfully ask whether we have addressed your questions and concerns adequately. We would be happy to clear up any additional questions.

---

> ### Author Response · Authors · 2023-11-23
> **[Reminder] Summarization of our rebuttal**
>
> Dear reviewer EEVN,
>
> We believe that we have addressed the concerns that you have raised. Specifically,
>
> 1. We have **clarified our paper's main contribution** that you mentioned above.
>
> 2. We have added an **efficiency comparison table** based on **training time** and **GPU memory consumption.**
>
> 3. We further **analyzed the additional layers** with different network architectures and **clarified the contribution of our refinement layers.**
>
> 4. We have **enhanced the editing experiment** by including **additional concepts**, such as **object insertion and style transfer.** Furthermore, we have introduced a comparison with **Instruct-NeRF2NeRF, considering mask constraints.**
>
> We would like to gently remind you that the end of the discussion period is imminent. We would appreciate it if you could let us know whether our comments addressed your concerns.
>
> Best regards,
> Authors

---

### Author Response · Authors · 2023-11-18
**General Reply to All Reviewers**

We sincerely thank all reviewers for the constructive feedback. Per all the reviewers' comments, we revised the manuscript. Here is the summary of changes to our paper, and reply for common questions.

- We included time and GPU memory comparison (efficiency comparison) results in our main paper section 4.
  - | **Metrics**       | **CLIP\-NeRF** | **NeRF\-Art** | **Instruct N2N** | Ours |
|-------------------|----------------|---------------|------------------|-----------|
| Fine\-tuning time | 6 min          | 15 min        | 90 min           | 14 min    |
| GPU Memory        | 17 GB          | 18 GB         | 15 GB            | 8 GB      |

  - We add the efficiency comparison with table to main section. we check the fine-tuning time and memory usage in Table. Among baselines, our method uses the lowest memory for training, with a much lower time compared to Instruct Nerf2Nerf. In the baselines of CLIP-Nerf and Nerf-Art, we experiment with using downsized images as higher resolution editing causes GPU memory overflow. For Instruct Nerf2Nerf, the fine-tuning process requires excessive time as it periodically translates the training images. Considering that our method shows outperforming quality in text-guided editing, our proposed scheme is efficient in both memory and time aspects. When comparing the time for the pre-training NeRF backbone model, we did not include a comparison since all baselines and ours take almost the same amount of time (about 10 minutes). More Details and comparison on pre-training time is in our Appendix.

- We included an anonymous project page link to show the free-view rendering video of our edited results. [https://ed-nerf.github.io/](https://ed-nerf.github.io/)
- We included experiments on instruct NeRF2NeRF with mask condition in Appendix D and Fig 7.
- We added an ablation study on architectures of refinement layers in Fig 6 and Appendix B
- We added an ablation study on using different mask generation methods in Fig 9 and Appendix E
- We added visualization of latent space features in Fig. 10 and Appendix F.
- We included style transfer results. in Fig 11.
- We included additional results on mixing DDS with SDS in Fig. 12
- We corrected typos in Fig.2( c), Eq.9, eq. 10, Eq.12, Eq13.

---

### Meta-Review · Area_Chair_6mmT · 2023-12-10

**Metareview:**

(a) This paper presents a novel 3D NeRF editing method, ED-NeRF, which integrates text-to-image diffusion models into 3D editing. The paper claims that by embedding real-world scenes into the latent space of a latent diffusion model (LDM) and utilizing a unique refinement layer, ED-NeRF offers faster training and enhanced editing capabilities compared to traditional NeRF editing methods. Additionally, it introduces an improved loss function, adapting the delta denoising score (DDS) distillation loss from 2D image editing to 3D, claiming superior performance over the standard score distillation sampling (SDS) loss.

(b) Strengths:
1. The embedding into LDM's latent space and the new refinement layer are proposed for more effective and efficient NeRF editing.
2. The adaptation of DDS for 3D editing is a good contribution, claimed to outperform existing SDS loss in editing tasks.
3. The paper presents experimental results showing the method's effectiveness in terms of speed and output quality.

(c) Weaknesses:
1. The presence of flickering artifacts in the free-view videos suggests issues with multi-view consistency, a crucial aspect of 3D NeRF editing. The submission could be strengthened by a more in-depth analysis of the multi-view consistency issue, particularly the causes and potential solutions for the flickering artifacts observed.

2. Some reviewers questioned the novelty in the integration of different techniques. A more detailed discussion on how ED-NeRF's approach differs fundamentally from existing methods would enhance the paper's value.

3. There were initial concerns about the method's generalizability and its sensitivity to the accuracy of object masks, which, though partially addressed, could benefit from further elaboration.

**Justification For Why Not Higher Score:**

1. The presence of flickering artifacts in the free-view videos suggests issues with multi-view consistency, a crucial aspect of 3D NeRF editing. The submission could be strengthened by a more in-depth analysis of the multi-view consistency issue, particularly the causes and potential solutions for the flickering artifacts observed.

2. Some reviewers questioned the novelty in the integration of different techniques. A more detailed discussion on how ED-NeRF's approach differs fundamentally from existing methods would enhance the paper's value.

3. There were initial concerns about the method's generalizability and its sensitivity to the accuracy of object masks, which, though partially addressed, could benefit from further elaboration.

**Justification For Why Not Lower Score:**

1. The embedding into LDM's latent space and the new refinement layer are proposed for more effective and efficient NeRF editing.
2. The adaptation of DDS for 3D editing is a good contribution, claimed to outperform existing SDS loss in editing tasks.
3. The paper presents experimental results showing the method's effectiveness in terms of speed and output quality.

---

### Decision · Program_Chairs · 2024-01-16

Accept (poster)